# Effects of Physical Self-Concept, Emotional Isolation, and Family Functioning on Attitudes towards Physical Education in Adolescents: Structural Equation Analysis

**DOI:** 10.3390/ijerph17010094

**Published:** 2019-12-21

**Authors:** Rosario Padial-Ruz, José Antonio Pérez-Turpin, Mar Cepero-González, Félix Zurita-Ortega

**Affiliations:** 1Department of Didactics of Musical, Plastic and Corporal Expression, University of Granada, 18071 Granada, Spain; mcepero@ugr.es (M.C.-G.); felixzo@ugr.es (F.Z.-O.); 2Department of General Didactics and Specific Didactics, University of Alicante, 03690 San Vicente del Raspeig, Alicante, Spain; jose.perez@ua.es

**Keywords:** self-concept, isolation, physical activity, family, adolescent

## Abstract

(1) Background: The present research seeks to define and contrast an explanatory model of physical self-concept, emotional isolation, attitude towards physical education, and family functioning, and analyse the existing associations between these variables. (2) Methods: The sample was made up of 2388 adolescents (43.39% male and 56.61% female), with ages of 11–17 years (M = 13.85; SD = 1.26) from Spain. Self-concept (AF-5), Isolation (UCLA), Attitude towards Physical Education (CAEF), and Family Functioning (APGAR) were analyzed. (3) Results: Good fit was obtained for all evaluation indices included in the structural equation model, which was significantly adjusted (χ^2^ = 233,023; DF = 14; *p* < 0.001; comparative fit index (CFI) = 0.913; normalized fit index (NFI) = 0.917; incremental fit index (IFI) = 0.906; root mean square error of approximation (RMSEA) = 0.072). (4) Conclusions: Attitudes towards physical activity were found to be positive when isolation levels were low and where adequate self-concept existed, specifically in students reporting high family functioning.

## 1. Introduction

Lack of physical activity (PA) in the 21st century, both in children and in adults, has become a topic of interest worldwide in developed countries. The increase in sedentary habits as a consequence of technological advances has provoked a rise in physical inactivity within the population. This has negative consequences for physical health and mental wellbeing [1,2]. This situation is reflected in the growth of both physical diseases such as obesity, diabetes, and cardiovascular diseases (with inactivity being considered the fourth leading cause of death in the world [3]), and mental illnesses [4,5], in addition to anti-social behaviors such as school bullying and violence [6].

PA during infancy and adolescence is associated with physical health, and psychological and social benefits over both the medium and long term [7]. Despite this, children do not currently meet recommendations laid out by the World Health Organisation for five to 17-year olds, of at least 60 min of moderate to vigorous PA a day [3]. This should be considered alongside evidence that more young people abandon sport when they begin the secondary school stage [8,9,10]. In order to tackle this problem, it is necessary to initiate prevention processes during the adolescent phase, given that the majority of risk factors related with sedentary behaviors start at this age [11]. It is during adolescence where the processes that lead to the construction of personality are consolidated. This is, therefore, a fundamental stage for its development and for the acquisition of values which facilitate positive interactions with others [12]. 

Physical education (PE) in the educational setting provides a tool for these learnings [8,12]. Amongst the different potential benefits of physical education within these age groups that are related to psychological and mental wellbeing, we find social development and the development of personality. Further, PE promotes fundamental behaviors and skills related to emotional intelligence such as self-concept, altruism, empathy, and understanding the needs of others, amongst others [13].

In adolescents, engaging in physical and sporting activity contributes to the promotion of a positive personality and emotional empathy, which improves interpersonal relationships between participants [14,15]. Levels of self-concept and empathy represent protective factors against the development of violent behaviors and victimization in scholars [16], whilst also improving positive relationships with others and mental health [17]. Self-concept is understood as a mental representation of the way in which individuals see themselves [18]. It is composed of diverse social factors—social, emotional, physical, family, and academic [19]. Research studies demonstrate that adolescents, both males and females, who participate regularly in PA report a more positive physical self-concept than those who are not as active [20]. This is the driver of improved social perception, which enables better relationships with others and leads to better academic performance [21].

Recent studies have added a new risk factor for mental health and anti-social conduct in adolescents. This is the rise in screen time [22], which together with the reduction in PA [23], drives an increase in feelings of loneliness. High self-efficacy towards the control and reduction of screen time and engagement in PA are factors that relate to a lower probability of social isolation and feelings of loneliness [24,25,26].

The school setting and, specifically, school physical activity, provides an ideal setting for the prevention of inactivity and promotion of PA. Nonetheless, schools by themselves cannot cover the daily recommendations that will bring an end to the health problems generated by inactivity; however, they can influence the acquisition of healthy habits [2,3]. Together with the school environment, family is another fundamental agent of socialization. Indeed, numerous studies conclude that both the mother and the father have a fundamental role in the PA engagement of their children [27,28], their motivation towards active pursuits [29], and as facilitators of their engagement in leisure-sport activities [30]. Recent studies also show the impact of the socio-economic level and mental health of parents, especially mothers. These contribute to developing healthier lifestyles through both the facilitation of PA and good nutrition, thus leading to lower indices of obesity amongst children [31]. The results show that social, family, and environmental factors become motivational elements for regular engagement in PA [32]. 

Due to the scarcity of studies that approach self-concept, loneliness, physical activity and family, the following study states the objectives of defining and contrasting an explanatory model of self-concept, isolation, physical activity, and family. In addition, it will analyze the existing associations between variables through multi-group structural equation analysis. 

## 2. Materials and Methods 

### 2.1. Design and Participants

The present research is descriptive and cross-sectional in nature. A total of 2388 adolescents participated who reported being aged between 11 and 17 years (M = 13.85 years; SD = 1.268). The sample included 1036 (43.39%) males and 1352 (56.61%) females. All participants were enrolled on the third year of primary school, or the 1st or 2nd year of compulsory secondary education (CSE) in Andalusia. Sample selection took place through a convenience sampling strategy, attending to the criteria of being enrolled on the third year of primary education, or the 1st or 2nd year of CSE. All participants had informed consent from their parents or legal guardians and did not suffer from any type of pathology that impeded their ability to participate in the research. This formed the inclusion and exclusion criteria. The sample was obtained from eight Spanish cities, with participation being requested from all centers who voluntarily agreed to participate. It is necessary to indicate that 281 questionnaires were excluded after it was detected that they had been incorrectly completed or had missing data. 

### 2.2. Variables and Instruments

Ad hoc questionnaire: For the selection of descriptive variables, various aspects were considered which could establish differences at some stage of the research process. These included sex, school year, population, engagement in PA outside of school hours, and the place in which individuals engage in PA. 

Self-concept Questionnaire: Data is collected through the original questionnaire “Autoconcepto Forma-5 [Self-concept Form] (AF-5)” of García and Musitu [19]. It measures the dimensions of Academic Self-concept (AA), Social Self-concept (AS), Emotional Self-concept (AE), Family Self-concept (AFM) and Physical Self-concept (AF). This test includes 30 questions which are rated along a five-point Likert scale, where 1 is never and 5 is always. 

In the study conducted by García and Musitu [19] reliability of α = 0.810 was determined. This value is almost identical to that detected in the present work (Cronbach alpha α = 0.833). The values produced for each dimension (AA: α = 0.773; AS: α = 0.702; AE: α = 0.697; AFM: α = 0.778; AF: α = 0.721) and the values produced in all of the groups were satisfactory, in the same way as has been presented in studies by Estévez, Martínez, and Musitu [33], and Cava et al. [34].

Loneliness Scale (UCLA): This scale is based on the original created by Russell, Peplau, and Cutrona [35], and the adapted version of Russell [36]. The adaptation to Spanish used here corresponds to that described by Expósito and Moya [37]. It contains 20 items and targets students ages 11 years and up. The original factor structure is formed by a factor that reports a general index of the perception of loneliness, divided into the following dimensions: loneliness; emotional loneliness; and subjective evaluation of the social network. The scale presents reliability coefficients that range between 0.74 and 0.94 depending on the population within which the questionnaire is administered [36,38], and shows adequate test-pre-test reliability [39]. Excellent psychometric properties have been observed in studies carried out with Spanish adolescents [40,41,42,43]. In the present research, the value obtained for the Cronbach alpha was 0.89. Coefficients of internal consistency for the bi-factor structure are as follows: Cronbach alpha of 0.84 and 0.83, respectively, and 0.88 for the complete scale.

Questionnaire on attitudes towards physical education (CAEF): The original questionnaire of Moreno, Rodríguez, and Gutiérrez [44] will be used. This comprises 56 items rated along a four-point Likert-type scale, where 1 = disagree and 4 = totally agree. This instrument is composed of seven different dimensions: Rating of the subject and of the PE teacher, PE difficulty, PE usefulness, empathy with the teacher and the subject, agreement with subject management, preference for PE and sport, and PE as sport. A consistency value of α = 0.75 was obtained for this instrument. This value is acceptable and slightly higher than the value obtained by Moreno et al. [44] in their original study (α = 0.73). 

Family Functioning Scale (APGAR): This test is extracted from the original version “Family APGAR” developed by Smilkstein, Ashworth, and Montano [45] and adapted to Spanish by Bellón, Delgado, Luna, and Lardelli [46]. It uses a three-point Likert scale (0 = almost never, 1 = sometimes and 2 = almost always), along which five positively-framed items are rated. It generates three types of functionality: severe dysfunction (D.S), moderate dysfunction (D.M), and family functioning (F.F). Internal consistency of the questionnaire in its original version is α = 0.750, whilst the authors Sánchez, Villarreal, and Musitu [47] more recently reported an internal consistency of α = 0.790. 

### 2.3. Procedure

Educational centers were contacted from the University of Granada in order to inform them about the nature of the study, with the centers that voluntarily agreed to participate then being selected to participate in the research. Informed consent packs were given out to students at the center, requesting collaboration from their parents or legal guardians. Next, questionnaires were administered to the group during lesson time. Anonymity of participants was guaranteed, clarifying that collected data would be used purely for scientific purposes. Researchers were present during data collection in order to guarantee the correct development of processes and to resolve any doubts. The present research study received approval from the Ethics Committee of the University of Granada with code641/CEIH/2018.

### 2.4. Data Analysis

The statistical software IBM SPSS^®^ in its version 23.0 (SPSS Inc., Chicago, IL, USA) for Windows was used for the analysis of basic descriptive data. The program IBM AMOS^®^ 23 (International Business Machines Corporation, Armonk, NY, USA)) was employed with the aim of analyzing the existing relationships between the constructs included in the structural model. After developing the theoretical model, a path analysis was carried out considering matrix associations through a multi-group analysis which grouped participants as a function of whether or not they regularly participated in physical activity. Finally, a path model was constructed which constituted nine factors (Figure 1). Difficulty of physical education (DEF), usefulness of physical education (UEF), empathy with the teacher (EPA), and agreement with subject management (COA) act as exogenous variables in the model. On the other hand, the variables describing rating of the subject or of the teacher (VPEF) and the preference of physical education as sport (PEFD) receive the effects of exogenous variables, whilst emotional isolation (SOLEM), family functioning (APGAR), and physical self-concept (AF) receive the effects of VPEF and PEFD. These last five variables act as endogenous variables within the model and its associated error terms. 

Unidirectional arrows show the effects between the variables incorporated (direct and indirect). Likewise, parameter estimation was carried out through the maximum likelihood method (ML) as this method is coherent, unbiased, and invariant to scale type. Error terms were established for all of the endogenous variables. 

Model fit was examined with the aim of verifying compatibility of the model to the empirical data obtained. Analysis of the reliability of model fit was performed according to goodness of fit criteria established by Marsh [48], p. 785. In the case of the chi-squared analysis, non-significant values associated top indicate good model fit. Comparative fit index (CFI) values will be considered acceptable if they are greater than 0.90 and excellent if they are greater than 0.95. The normalized fit index (NFI) should not be greater than 0.90. Incremental fit index (IFI) values will be considered acceptable if they are greater than 0.90 and excellent if they are greater than 0.95. Finally, root mean square error of approximation (RMSEA) values will be considered excellent if they are lower than 0.05 and acceptable if they are lower than 0.08. 

## 3. Results

With regards to the descriptive data referred to in Table 1, the sample is composed of a total of 2388 students of both sexes (48.2% males and 51.8% females), enrolled on the 3rd year of primary education or the 1st year of CSE in one of the eight provinces of Andalusia. 81.5% of participants belong to a functional family, 15.1% to a moderately functional family and 3.4% to families with signs of serious dysfunction. 

With regards to attitudes towards physical education, the most highly rated dimension is agreement with EF management (M = 3.01). All of the other dimensions were found to have values lower than 3, with the lowest mean pertaining to usefulness of PE (M = 2.06).

With regards to self-concept, the academic dimension received the highest rating (M = 3.61), followed by the physical (M = 3.54), social (M = 3.48) and family (M = 3.39) dimensions, and finally, emotional self-concept (M = 3.02). In reference to the variable describing isolation, the highest score was achieved for the dimension describing subjective evaluations of the social network (M = 2.94), followed by general isolation (M = 2.00), and finally, emotional isolation (M = 1.95).

Table 2 shows the mean scores obtained for the dimensions of attitude towards physical education and family functioning, finding statistically significant differences (*p* = 0.000). These are reflected in higher ratings of the subject and of the teacher on behalf of students who have highly functioning families (M = 2.76), relative to those who have moderately dysfunctional families (M = 2.59). The dimension describing difficulty of PE (*p* = 0.010) is more highly scored by students who report moderate family dysfunction (M = 2.49) and scored less highly by those with families with severe dysfunction (M = 2.32). With respect to the usefulness of PE, this dimension is more highly scored by students with families with moderate dysfunction (M = 2.21), in comparison to those with highly functional families (M = 2.02). Those who proportion higher scores to the dimension of empathy with the teacher and subject (M = 2.45), and agreement with subject management (M = 3.03) also reported good family functioning. Further, for preference for PE and sport (*p* = 0.005), the highest score was also obtained for those who have high family functioning (M = 2.33) relative to serious dysfunction (M = 2.10).

Table 3 shows the mean scores obtained for the dimensions of self-concept in relation to family functioning. In the following table it can be seen that academic self-concept (*p* = 0.000 ***) reflects higher scores for those who have a highly functional family (M = 3.69), with scores being lower when family dysfunction is serious (M = 3.06). 

Social self-concept reaches higher scores amongst those with highly functioning families (M = 3.54) relative to those who present serious (M = 3.10) or moderate dysfunction (M = 3.26). However, in the case of emotional self-concept the highest scores are obtained for those from families with moderate dysfunction (M = 3.12) and the lowest scores relate to those with serious dysfunction (M = 2.99). Finally, it must be highlighted that both family self-concept (M = 3.45) and physical self-concept (M = 3.62) achieve their highest mean values in students from highly functional families, this value being lowest in students from families with serious dysfunction (M < 3).

Finally, with regards to the dimensions of isolation in relation to family functioning, statistically significant differences were achieved (*p* = 0.000 ***) in all of its dimensions (Table 4). Students with moderately dysfunctional families reflect higher levels of emotional isolation (M = 2.36) relative to those who have highly functional families (M = 1.89). In contrast, higher scores are obtained for subjective evaluations of the social network in students with highly functional families (M = 3.00) relative to those who have families with serious dysfunction (M = 2.54). This is in contrast to general isolation whose highest mean value pertained to serious family dysfunction (M = 2.401) and lowest value pertained to those who were highly functional (M = 1.93).

Table 5 shows the bivariate correlations between study variables. Turning attention to the physical dimension of self-concept in relation to attitudes towards PE, a positive correlation is observed with the dimensions describing students’ ratings of the subject of physical education and its teacher (*r* = 0.230 **), empathy towards the teacher and subject *r* = 0.229 **), agreement with subject management (*r* = 0.210 **) and preference for PE and sport (*r* = 0.236 **).The positive correlation is weakest in relation to the dimensions describing the difficulty presented by PE (*r* = 0.153 **) and PE as sport (*r* = 0.060 **), whilst a low negative correlation is seen with usefulness of EF(*r* = −0.046 *).

Attending to the dimension of emotional isolation in relation to attitudes towards Physical Education, it is observed that when emotional isolation is higher, ratings of the subject and the teacher imparting it are lower (*r* = −0.170 **). In contrast, when this emotional isolation is greater, perceived difficulty of this subject is also higher (*r* = 0.066 **). In the same way, a medium correlation is produced between the dimension of isolation and usefulness of PE (*r* = 0.323 **), however, when this emotional isolation is greater, agreement with subject management decreases (*r* = −0.191 **). 

In addition, a slight agreement is maintained regarding preference for PE and sport (*r* = 0.107 **), and with PE as sport (*r* = 0.085 **).

Once the descriptive and comparative analysis had been performed, a structural equation model was constructed (Figure 2) which included the questionnaire items pertaining to attitude towards PE together with the variables related to it: emotional isolation (SOLEM), family functioning (APGAR), and physical self-concept (AF). 

Good fit was obtained for all evaluation indices of the structural equation model. Chi-squared analysis revealed a significant *p*-value p (χ^2^ = 233.023; df = 14; *p* < 0.001), although we must bear in mind that this statistic, as an index, has no upper limit. Further, problems arise as it cannot be interpreted in a standardized way and is sensitive to sample size. In addressing this, other indices of standardized fit are employed which are less sensitive to sample size. The comparative fit index (CFI) showed a value of 0.913, this being acceptable. The normalized fit index (NFI) specified a value of 0.917 and the incremental fit index (IFI) was 0.906, both being acceptable. Analysis of the root mean square error of approximation (RMSEA) obtained an acceptable value of 0.072.

Table 6 and Figure 2 present the values produced for the associations between the variables included in the structural equation model developed. Addressing the first level of the model (associations found between DEF, UEF, EPA and COA), all of the exogenous variables of the model show statistically significant relationships at the level *p* < 0.001. These were positive and direct in all cases except for the association between COA and UEF (*r* = −0.110). 

Approaching the second level, it can be observed that VPEF is positively associated with DEF (*r* = 0.127; *p* < 0.001), COA (*r* = 0.230; *p* < 0.001), and EPA (*r* = 0.309; *p* < 0.001), whilst the association between VPEF and UEF was negative (*r* = 0.170; *p* < 0.001). On the other hand, if the associations between PEFD and the exogenous variables are considered, statistically significant relationships are shown with DEF (*r* = 0.054; *p* < 0.01), UEF (*r* = 0.204; *p* < 0,001), and EPA (*r* = 0.335; *p* < 0.001). No statistically significant associations were obtained between COA and PEFD (*p* = 0.544). 

Following this, the endogenous variables in the second level of the model are related with emotional isolation, family functioning, and physical self-concept. In the first instance, a positive relationship can be observed between VPEF and family functioning (*r* = 0.143; *p* < 0.001), and between VPEF and physical self-concept (*r* = 0.154; *p* < 0.001), whilst this variable was negatively related with emotional isolation (*r* = −0.177; *p* < 0.001). Along similar lines, PEFD was directly associated with physical self-concept (*r* = 0.189; *p* < 0.001) and emotional isolation (*r* = 0.161; *p* < 0.001), although in this case an association was not found with family functioning (*p* = 0.544). 

Finally, it is highlighted that family functioning maintains a positive and direct relationship with physical self-concept (*r* = 0.207; *p* < 0.001), whilst this association was negative and indirect with emotional isolation (*r* = −0.224; *p* < 0.001). 

## 4. Discussion

The present study, conducted with students in the 3rd year of primary education or the 1st year of CSE from the eight provinces of Andalusia, pursues as its principal objectives the analysis of existing relationships between the attitudes towards physical education of schoolchildren and their physical self-concept, level of emotional isolation and family functioning. These premises are of great interest when it comes to developing and adapting concrete teaching methods and strategies, which favor the development of programs that incentivize students’ participation in AF and assume a positive role in the psychosocial development of students [49]. 

Analyzing the family context, it is deduced that eight out of every ten students belong to a family with good family functioning, with very few students coming from families with severe dysfunction. This type of family functioning will facilitate engagement in PA [50], although some authors maintain that students reporting dysfunctional family contexts can report high levels of PA, suggesting that PA may exert a compensatory effect [51]. Further, a good family climate will serve to extrapolate this effect to the social relationships maintained within individuals’ personal context. This not only relates to their affect for and trust in others, but also to the way in which they approach conflict resolution or help others. 

In reference to attitudes towards PE, the most highly rated dimension is agreement with subject management relating to PE [52], in contrast to the usefulness of this subject which demonstrated a trend towards considering PE as of little use [53,54]. On the contrary, numerous studies positively rate the usefulness of engaging in sport and PA on a daily basis [55,56]. This positive evaluation of the usefulness of PE leads to intrinsic motivation for engaging in PA, both inside and outside of the school timetable [57].

Another significant fact uncovered by the study is that self-concept, in all of its dimensions, achieved a higher level amongst individuals who engaged in PA. This confirms the positive effects of PA at a physical and mental level, on social relations, and on academic performance [13], this being even more evident during the adolescent stage [58]. Other studies carried out on self-concept and PE, relate engagement in PA with improved physical self-concept in a more binding way, whilst failing to find relationships with other dimensions of self-concept [59]. Higher levels of isolation, both in a general and in an emotional sense, are evident within students who do not engage in PA [60], with subjective evaluations of the social network being greater amongst those who are active [61]. However, other authors maintain that greater subjective evaluations of the social network come from students who do not count on an excessive number of social links, given that their subjective social network is smaller and can be more effectively controlled [62].

With regards to family relations and engagement in PA, data from both the present study and other research studies indicate that the level of family functioning does not have a significant relationship with engagement in PA and/or sport [63]. Nevertheless, studies such as those conducted by Aaltonen, Kaprio, Kujala, Pulkkinen, Rose, and Silventoinen [64] argue that when family functioning is good, so are the PA levels of family members. Attitudes towards PE and family functioning, students who rate the subject and teacher more highly, report agreement with subject management and state a preference for EF and sport, also tend to present high family functioning. This is the case despite these students considering the subject to be of little use [65]. In contrast, those who rate the teacher and physical education less highly tend to be those students with moderately dysfunctional families, with these students also perceiving the aforementioned subject as being more challenging yet highly useful [66]. Thus, lack of attention to the basic psychological needs from within the family unit and positive relations between members, will have negative social repercussions for these students at school. Such repercussions could impinge upon their attitudes and interest towards PA and sport engagement.

Good family functioning is also related with a high academic and social self-concept [67], in contrast to the higher emotional self-concept seen in students with families that present moderate dysfunction [68]. Both family and physical self-concept are higher in students who belong to functional families, whilst lower levels correspond to families with a serious dysfunction [69]. However, other research studies subscribe to different conceptions. These have demonstrated that family self-concept is higher within students with functional families, whilst no direct link was established between physical self-concept and family type, but was with the development of abilities through sporting practice [70]. Other works do not link the level of self-concept with the type of family to which students belong [71].

With respect to the variables of isolation according to family functioning, we found that emotional isolation predominates above all within those students who belong to severely dysfunctional families. High scores are obtained within functional families for the subjective evaluation of the social network, with this instead being low within those who live within a seriously dysfunctional family. In the same way, students who present greater general isolation are those who belong to seriously dysfunctional families. Research conducted by other authors collaborate these results [72], whilst others such as Twenge, Spitzburg, and Campbell [73] did not manage to demonstrate that isolation in adolescents is more linked to one type of family over another type. These authors instead found high levels of isolation within adolescents coming from functional families. This makes it clear that their isolation could be related to a lack of activities performed with peers during out of school hours and not to family typology.

Analysis of bivariate correlations performed with the variables of physical self-concept, emotional isolation and attitude towards PE, demonstrated that students who have a good physical self-concept present scant levels of isolation at a general or emotional level. These data are in agreement with data produced by other studies and reflect that the lower the level of isolation in students, the greater their physical self-concept and security of the social relationships they establish with their social environment. All of this facilitates greater engagement in PA [74]. In the same way, greater physical self-concept is generally related with stronger attitudes towards PE [75,76], with a positive relationship between preference for PE and sport, and a negative relationship with its usefulness standing out.

Attending to the dimensions of isolation in relation to those pertaining to attitudes towards physical education, it is observed that when emotional isolation increases in students, the rating that they attribute to the subject and teacher is lower [77,78], and they report less agreement with respect to subject management [79,80]. However, we demonstrate that this is not a specific aspect directed towards the teacher or this subject but is a trait that these types of students maintain towards teaching staff and all subjects generally speaking [81]. Nevertheless, they do perceive this subject to be more difficult [80] and highly useful [79]. This is in contrast to findings reported in other studies such as that conducted by Krause, Gulick, and Basin [82].

With regards to the dimensions pertaining to attitude towards PE, students who rate the subject as more difficult also tend to perceive it as being more useful [83]. Those who rate the subject and teacher more highly also demonstrate greater agreement with its management [84], whilst in contrast, also considering it to be less useful. When students conceive PE as sport, they find it more difficult as a subject [85], view it as being more useful, and hold a stronger preference for it [86].

From the structural equations carried out it can be seen that an increase in the perceived difficulty of physical education goes hand in hand with an increase in reports of its usefulness, coherence in the way it is managed and empathy towards the teacher who imparts it [87]. It can be confirmed that when each one of these variables increases, so to do the other variables, except in the case of coherence in subject management. Instead, when this increases, subject usefulness decreases. For its part, it is observed that ratings of the subject of PE improve when the difficulty of PE, coherence, organization, and empathy for the teacher increases. This final point has also been endorsed in other studies [88]. However, ratings of PE decrease when the usefulness of PE increases [89]. In the same way, ratings of the subject of PE improve when the difficulty of PE, coherence and organization increase. However, just as has been stated by other authors [90], ratings of PE decrease when the usefulness of PE increases. Finally, it is of note that preferences for PE and sport increase when met by increases in the perceived difficulty and usefulness of PE.

This study presents some limitations. The first of these is due to the fact that it deals with a cross-sectional study and so is not able to establish causal relationships. A second limitation is that this study does not enable generalization of the data obtained to other populations. This being said, the explanatory model developed does enable a better understanding of the associations between variables. Further, we consider the sample to be broad and representative of the targeted population. For this reason, the present research permits development of a future line of research, providing a base from which we can evaluate, compare, and replicate the study with other groups of students. Findings will also be used for the planning and design of future research in a classroom setting.

## 5. Conclusions

The main conclusions of the present study confirm that the physical dimension of self-concept is high in students who maintain low percentages in all of the dimensions of isolation. Those families in which functionality is good engage regularly in AF. At the same time, it is also reflected that students who rate the subject and teacher more highly, whilst also demonstrating empathy towards both, tend to be those who present relatively high family functioning. The same can be seen to occur in reference to subject management, and preference for PE and sport. With regards to the theoretical model of self-concept, isolation, family functioning and attitude towards PE, it presented good fit. The data indicate that engagement in PA is not determined by the other variables measured in the model. Results also suggest that family functioning intervenes to a large extent on self-concept and its dimensions, in addition to the isolation demonstrated by students. It is concluded that attitudes towards PA are positive when levels of isolation are low and when an adequate self-concept is present. All of this is typically generated in students who, in the majority of cases, come from families with high levels of functioning.

## Figures and Tables

**Figure 1 ijerph-17-00094-f001:**
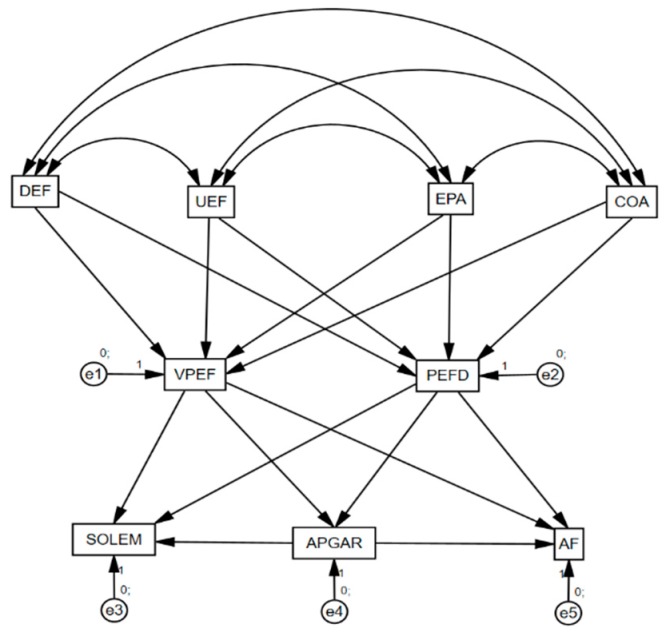
Analysis of data. Note 1: Difficulty of physical education (DEF); usefulness of physical education (UEF); empathy with the teacher (EPA); agreement with subject management (COA); rating of the subject and the teacher (VPEF); preference of physical education as sport (PEFD); emotional isolation (SOLEM); family functioning (APGAR); physical self-concept (AF).

**Figure 2 ijerph-17-00094-f002:**
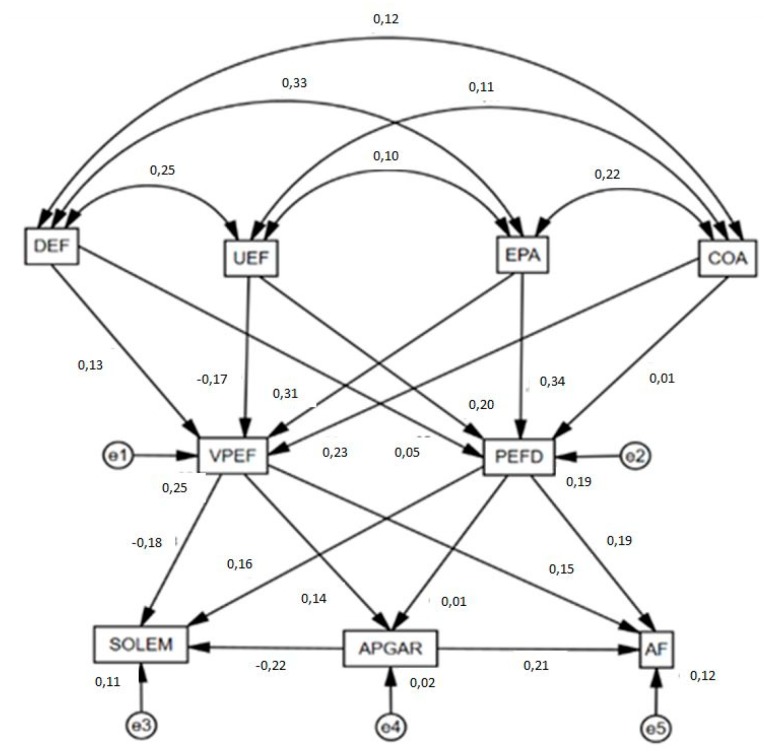
Structural equation model. Note 1: Difficulty of physical education (DEF); usefulness of physical education (UEF); empathy with the teacher (EPA); agreement with subject management (COA); rating of the subject and teacher (VPEF); preference for physical education as sport (PEFD); emotional isolation (SOLEM); family functioning (APGAR); physical self-concept (AF).

**Table 1 ijerph-17-00094-t001:** Descriptive analysis of variables.

Gender		Isolation	
Male	48.2% (*n* = 1151)	General isolation	M = 2.00 (SD = 0.505)
Female	51.8% (*n* = 1237)	Subjective evaluation of social network	M = 2.94 (SD = 0.571)
**Family functioning**		Emotional isolation	M = 1.95 (SD = 0.574)
Serious dysfunction	3.4% (*n* = 82)	**Attitude towards physical education**	
Moderately dysfunctional family	15.1% (*n* = 360)	Rating of the subject and teacher of PE	M = 2.73 (SD = 0.521)
Highly functional family	81.5%(*n* = 1.946)	Difficulty of PE	M = 2.42 (SD = 0.546)
**Self-concept**		Usefulness of PE	M = 2.06 (SD = 0.442)
Academic self-concept	M = 3.61 (SD = 0.821)	Empathy with the teacher and subject	M = 2.44 (SD = 0.613)
Social self-concept	M = 3.48 (SD = 0.574)	Preference for PE and sport	M = 2.32 (SD = 0.618)
Emotional self-concept	M = 3.02 (SD = 0.765)	PE as sport	M = 2.45 (SD = 0.680)
Family self-concept	M = 3.39 (SD = 0.508)	Rating of the subject and teacher of PE	M = 2.73 (SD = 0.521)
Physical self-concept	M = 3.54 (SD = 0.757)		

Mean (M); Standard Deviation (SD).

**Table 2 ijerph-17-00094-t002:** Between attitude towards physical education and family functioning.

Dimensions		Mean	Standard Deviation	F	Sig
**Rating of the subject and teacher of PE**	Serious dysfunction	2.46	0.568	26.191	0.000 ***
	Moderate dysfunction	2.59	0.495		
	Highly functional family	2.76	0.517		
**Difficulty of PE**	Serious dysfunction	2.32	0.589	4.629	0.010 ***
	Moderate dysfunction	2.49	0.554		
	Highly functional family	2.41	0.541		
**Usefulness of PE**	Serious dysfunction	2.09	0.487	27.949	0.000 ***
	Moderate dysfunction	2.21	0.458		
	Highly functional family	2.02	0.430		
**Empathy with the teacher and subject**	Serious dysfunction	2.18	0.818	8.279	0.000 ***
	Moderate dysfunctional	2.40	0.619		
	Highly functional family	2.45	0.599		
**Agreement with subject management**	Serious dysfunction	2.91	0.658	7.460	0.001 ***
	Moderate dysfunction	2.90	0.603		
	Highly functional family	3.03	0.600		
**Preference for PE and sport**	Serious dysfunction	2.10	0.719	5.275	0.005 ***
	Moderate dysfunction	2.32	0.608		
	Highly functional family	2.33	0.613		
**PE as sport**	Serious dysfunction	2.47	0.770	0.325	0.723
	Moderate dysfunction	2.47	0.643		
	Highly functional family	2.44	0.682		

*** Statistically significant association between variables at the level 0.001.

**Table 3 ijerph-17-00094-t003:** Between self-concept and family functioning.

Dimensions		Mean	Standard Deviation	F	Sig
**Academic self-concept**	Serious dysfunction	3.06	0.899	64.459	0.000 ***
	Moderate dysfunction	3.26	0.776		
	Highly functional family	3.69	0.800		
**Social self-concept**	Serious dysfunction	3.10	0.680	57.132	0.000 ***
	Moderate dysfunction	3.26	0.622		
	Highly functional family	3.54	0.544		
**Emotional self-concept**	Serious dysfunction	2.99	0.878	3.712	0.025 ***
	Moderate dysfunction	3.12	0.725		
	Highly functional family	3.00	0.766		
**Family self-concept**	Serious dysfunction	2.94	0.568	71.045	0.000 ***
	Moderate dysfunction	3.21	0.546		
	Highly functional family	3.45	0.480		
**Physical self-concept**	Serious dysfunction	2.94	0.877	71.884	0.000 ***
	Moderate dysfunction	3.23	0.736		
	Highly functional family	3.62	0.725		

*** Statistically significant association between variables at the level 0.001.

**Table 4 ijerph-17-00094-t004:** Relation between isolation and family functioning.

Dimensions		Mean	Standard Deviation	F	Sig
**Emotional isolation**	Serious dysfunction	2.36	0.633	78.117	0.000 ***
	Moderate dysfunction	2.23	0.557		
	Highly functional family	1.89	0.553		
**Subjective evaluation of the social network**	Serious dysfunction	2.54	0.627	70.950	0.000 ***
	Moderate dysfunction	2.69	0.541		
	Highly functional family	3.00	0.554		
**General isolation**	Serious dysfunction	2.40	0.529	8.110	0.000 ***
	Moderate dysfunction	2.26	0.453		
	Highly functional family	1.93	0.489		

*** Statistically significant association between variables at the level 0.001.

**Table 5 ijerph-17-00094-t005:** Toward physical education in relation to physical self-concept and emotional isolation.

	Physical Self-Concept	Emotional Isolation	Rating of the Subject and Teacher of PE	Difficulty of PE	Usefulness of PE	Empathy with the Teacher and Subject	Agreement with Subject Management	Preference for PE and Sport
**Physical self-concept**	1							
**Emotional isolation**	−0.255 **	1						
**Rating of the subject and teacher of PE**	0.230 **	−0.170 **	1					
**Difficulty of PE**	0.153 **	0.066 **	0.212 **	1				
**Usefulness of PE**	−0.046 *	0.323 **	−0.133 **	0.253 **	1			
**Empathy with the teacher and subject**	0.229 **	0.000	0.385 **	0.326 **	0.100 **	1		
**Agreement with subject management**	0.210 **	−0.191 **	0.334 **	0.121 **	−0.110 **	0.224 **	1	
**Preference for PE and sport**	0.236 **	0.107 **	0.251 **	0.216 **	0.250 **	0.376 **	0.071 **	1
**PE as sport**	0.060 **	0.085 **	0.106 **	0.285 **	0.253 **	0.229 **	0.074 **	0.233 **

** Correlation significant at the level of 0.01; * Correlation significant at the level of 0.05.

**Table 6 ijerph-17-00094-t006:** Weights and standardized regression weights.

Associations between Variables			RW				SRW
			EST	S.E.	C.R.	*p*	EST
**PEFD**	←	DEF	0.061	0.023	2.671	**	0.054
**VPEF**	←	UEF	−0.201	0.022	−9.162	***	−0.170
**PEFD**	←	EPA	0.338	0.020	16.837	***	0.335
**PEFD**	←	COA	0.012	0.020	0.606	0.544	0.012
**VPEF**	←	DEF	0.121	0.019	6.516	***	0.127
**VPEF**	←	COA	0.199	0.016	12.462	***	0.230
**VPEF**	←	EPA	0.263	0.016	16.079	***	0.309
**PEFD**	←	UEF	0.285	0.027	10.570	***	0.204
**APGAR**	←	PEFD	0.010	0.016	0.607	0.544	0.012
**APGAR**	←	VPEF	0.135	0.019	7.027	***	0.143
**SOLEM**	←	VPEF	−0.196	0.022	−9.010	***	−0.177
**AF**	←	APGAR	0.319	0.030	10.683	***	0.207
**SOLEM**	←	APGAR	−0.264	0.023	−11.479	***	−0.224
**SOLEM**	←	PEFD	0.151	0.018	8.298	***	0.161
**AF**	←	VPEF	0.222	0.028	7.861	***	0.154
**AF**	←	PEFD	0.230	0.024	9.753	***	0.189
**COA**	↔	DEF	0.040	0.007	5.887	***	0.121
**DEF**	↔	EPA	0.109	0.007	15.130	***	0.326
**DEF**	↔	UEF	0.061	0.005	11.973	***	0.253
**EPA**	↔	UEF	0.027	0.006	4.881	***	0.100
**COA**	↔	UEF	−0.029	0.005	−5.361	***	−0.110
**COA**	↔	EPA	0.083	0.008	10.693	***	0.224

Note 1: RW, regression weights; SRW, standardized regression weights; EST, estimations; SE, standard error; CR, critical ratio. Note 2: Difficulty of physical education (DEF); usefulness of physical education (UEF); empathy with the teacher (EPA); agreement with subject management (COA); rating of the subject and teacher (VPEF); preference for physical education as sport (PEFD); emotional isolation (SOLEM); family functioning (APGAR); physical self-concept (AF). Note 3: * Statistically significant association between variables at the level 0.05; ** Statistically significant association between variables at the level 0.01, *** Statistically significant association between variables at the level 0.001.

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
