# Peer review of "Effects of Physical Self-Concept, Emotional Isolation, and Family Functioning on Attitudes towards Physical Education in Adolescents: Structural Equation Analysis"

_ijerph, 2019, doi:10.3390/ijerph17010094_

Round 1

Reviewer 1 Report

Dear Authors:
The article provides an interesting contribution to the field of physical activity and sport in primary school children. However, there is a need for improvement in a number of areas which are mentioned below:

The meaning of the acronym CSE should be explained in line 87 and included in parentheses after Andalusia (Spain). In the section on variables and instruments, there is no reference to whether any sociodemographic question or any question of interest for the study has been consulted. The presentation and format of Table 1 in the results should be improved and a note should be included in the table explaining the meaning of the acronyms SD (Standard Deviation) and M (Mean). A confirmatory factorial analysis of the different constructs used in the PATH is missing, as well as a measure of Composite Reliability and Average Variance Extracted (AVE). The results should be organized in different sections, for example:
1. Descriptive results.
2. Influence of family functioning
3. … Table 5 should be marked with a note indicating the meaning of ** and its level of significance. For example: Note: ** indicates statistically significant correlation p<.01 The RMSEA confidence interval (CI) should be reported in line 289. The correlation statistic (r) is used to report regression coefficients whose statistic is β The format of the note in table 6 should be revised as it should follow the formatting style of the journal template.
In the diagram of the PATH (figure 3) it is recommended to mark with an * the Beta coefficients that are significant and to indicate in a note of the figure the meaning of *. Also, as a suggestion, it is recommended to put with discontinuous line those relations that are not significant to improve the visualization and interpretation of the diagram. Finally, there is a lack of a section on practical implications and future lines of research, as well as indicating possible limitations of the research.

Reviewer 2 Report

This study employs an explanatory model of several factors on attitudes towards physical education in adolescents using a structural equation analysis. Overall, the study is well written and conducted. I must acknowledge that the statistical analysis approach is not a strength of mine but it appears rigorous. I have a few comments below to improve the manuscript for publication. 

Positives 

Significant sample size with appropriate validated measures. The outcome variables are particularly well defined (in particular the psychometric properties of the study instruments) I like that the authors have acknowledged that the study is descriptive, cross-sectional and exploratory. The structure of the paper is generally very good and clear.

Improvements (Major)

The study should undergo a full English language check (in particular in the introduction) to improve clarity. Some of the paragraphs are very short (2 sentences). This should be amended to improve the overall flow. Examples Ln 50-54, Ln 353-363, Ln 404-409. There is also one very very long paragraph Ln 369-387. Limitations should be more explicitly acknowledged around the cross-sectional nature of the study design and areas for future research articulated in the discussion. All Tables need to be redrawn and better presented. These look like screen shots from SPSS. These should be significantly improved with appropriate Figure legends.

Minor

Ln 136 space needed between research.Informed. Ln 323 (second sentence of discussion). Physical education/activity should be mentioned to better link teaching methods and strategies with this.

Round 2

Reviewer 2 Report

I recommend acceptance - thank you for addressing my comments